# Anti-Inflammatory Properties of the SGLT2 Inhibitor Empagliflozin in Activated Primary Microglia

**DOI:** 10.3390/cells11193107

**Published:** 2022-10-02

**Authors:** Marvin Heimke, Florian Lenz, Uta Rickert, Ralph Lucius, François Cossais

**Affiliations:** Institute of Anatomy, Kiel University, D-24118 Kiel, Germany

**Keywords:** microglia, neuroinflammation, neurodegeneration, empagliflozin, IL6, TNF, ERK1/2, NHE-1, SGLT1, SGLT2

## Abstract

Sodium-glucose cotransporter 2 (SGLT2) inhibitors, including empagliflozin, are routinely used as antidiabetic drugs. Recent studies indicate that beside its beneficial effects on blood glucose level, empagliflozin may also exert vascular anti-inflammatory and neuroprotective properties. In the brain, microglia are crucial mediators of inflammation, and neuroinflammation plays a key role in neurodegenerative disorders. Dampening microglia-mediated inflammation may slow down disease progression. In this context, we investigated the immunomodulatory effect of empagliflozin on activated primary microglia. As a validated experimental model, rat primary microglial cells were activated into a pro-inflammatory state by stimulation with LPS. The influence of empagliflozin on the expression of pro-inflammatory mediators (NO, *Nos2*, IL6, TNF, IL1B) and on the anti-inflammatory mediator IL10 was assessed using quantitative PCR and ELISA. Further, we investigated changes in the activation of the ERK1/2 cascade by Western blot and NFkB translocation by immunostaining. We observed that empagliflozin reduces the expression of pro- and anti-inflammatory mediators in LPS-activated primary microglia. These effects might be mediated by NHE-1, rather than by SGLT2, and by the further inhibition of the ERK1/2 and NFkB pathways. Our results support putative anti-inflammatory effects of empagliflozin on microglia and suggest that SGLT2 inhibitors may exert beneficial effects in neurodegenerative disorders.

## 1. Introduction

Neurodegenerative disorders, including Parkinson’s disease (PD) and Alzheimer’s disease (AD), represent a wide spectrum of pathologies. Despite their heterogeneity, neuroinflammation and reactivity of microglial cells represent a hallmark in the pathological processes associated with neurodegeneration. Whether the inflammatory reaction is the cause or consequence of neuronal loss is still not known [1]. Nonetheless, since inflammation sustains the pathological state, it was suggested that its containment may slow down the progression of neurodegenerative processes [2,3]. Microglial cells represent the resident macrophages of the CNS, permanently scanning the tissue for pathogens, tissue damages and abnormal endogenous protein aggregates [4,5,6]. After activation, they initiate an immune response through the release of pro-inflammatory mediators such as NO (via upregulated expression of NOS2), IL6, TNF and IL1B [5,6]. The immune response should be temporary, as prolonged inflammation may also harm neurons [7]. At the late onset of the inflammatory reaction, microglia usually pass into a neuroprotective phenotype, which is characterized by the release of anti-inflammatory cytokines, such as IL10, and growth factors [5]. This state is indispensable in the context of tissue healing [8,9]. Although this model might be over-simplified, the balance between a pro- and anti-inflammatory phenotype has been proposed to be disturbed in neurodegenerative diseases [10].

Similarly to neurodegenerative diseases, the prevalence of type 2 diabetes (T2DM) is constantly increasing worldwide. Inhibitors of the sodium/glucose cotransporter SGLT2 (solute carrier family 5 member 2, SLC5A2) have recently arisen as therapeutic agents for the treatment of T2DM. In this context, SGLT2 inhibitors effectively reduce blood glucose levels through increased glucosuria and decreased glucose toxicity [11,12]. Empagliflozin is one of the main SGLT2 inhibitors routinely used in the treatment of T2DM [13]. Besides its functions in the kidney, empagliflozin exerts beneficial cardiovascular effects and reduces oxidative stress [14,15]. Recent studies have indicated that empagliflozin might also mediate neuroprotective effects [16]. For instance, empagliflozin has been proposed to reduce vascular inflammation and to improve cognitive impairment in a mixed mouse model for T2DM and AD [17]. Lin et al. observed a reduction in oxidative stress markers in the CNS of db/db mice and an increased release of brain derived neurotrophic factor (BDNF) after treatment with empagliflozin, which was associated with cognitive improvement [18]. Hierro-Bujalance et al. observed reduced levels of β-amyloid and decreased densities of senile plaques in the brains of AD mice after treatment with empagliflozin. This effect was associated with an improvement of microglia burden in the CNS [17].

Together, these data suggest that empagliflozin may be beneficial for patients presenting with neurodegenerative disorders. However, to date, little is known regarding the cellular effects of SGLT2 inhibitors on microglia activation in the context of inflammation. In this work, we investigated the immunomodulatory properties of empagliflozin in an LPS-induced model of inflammation on rat primary microglia.

## 2. Materials and Methods

### 2.1. Culture of Primary Microglial Cells

Microglial cells were prepared from the rostral mesencephali and cerebral cortex of 1-to-3-day old Sprague Dawley and Wistar rats as previously described [6]. After decapitation, both hemispheres and the mesencephalon were isolated, minced and enzymatically dissociated with trypsin. The cell suspension was transferred to 75 cm^2^ cell culture flasks and incubated for 9 to 10 days in a humidified atmosphere with 5% CO_2_. The cell culture medium, consisting of Dulbecco’s Modified Eagle Medium (DMEM, Invitrogen, Karlsruhe, Germany), deactivated fetal calf serum (FCS, Invitrogen), penicillin (PAN Biotech, Aidenbach, Germany), streptomycin (PAN Biotech) and L-glutamine (PAN Biotech), was replaced every 48 h. After incubation, the microglial cells were harvested by shaking and decantation of the free-floating cells in the supernatant. Trypan blue exclusion was performed to estimate the number of viable cells. All experiments were performed in agreement with the local Ethics Committee (V312-7224.121-4) and in accordance with the 3R principles (Replacement, Reduction and Refinement) to reduce the number of animals sacrificed at our institute.

### 2.2. Cell Stimulation

Empagliflozin (Jardiance, Boehringer-Ingelheim, Ingelheim am Rhein, Germany) tablets were crushed mechanically and dissolved in 1 mL stimulation medium, consisting of DMEM (Invitrogen), deactivated FCS (Invitrogen), penicillin (PAN Biotech) and streptomycin (PAN Biotech), at a concentration of 1 mM. To improve the solubility, 1 µL Dimethylsulfoxid was added. For the induction of microglial activation, lipopolysaccharide (LPS) from salmonella typhimurium (Sigma-Aldrich, Taufkirchen, Germany) was used. After seeding the microglial cells at a given amount for 24 h, the cells were stimulated either with empagliflozin (0.5 µM, 1 µM, 5 µM and 50 µM for MTT assay and 50 µM for further experiments) or with LPS (5 ng/mL), or with a combination of both. For all co-treatments, the microglial cells were pre-treated with empagliflozin in the stated concentration for 30 min, before adding LPS for further co-stimulation with both substances. The stimulation time was as follows: 30 min for Western blot, 30 min for immunofluorescence staining and 24 h for cytotoxicity tests, measurement of nitric oxide production, qPCR and ELISA.

### 2.3. Cytotoxicity Test

The cell viability of microglia under stimulation with empagliflozin was monitored by methylthiazolyldiphenyl-tetrazolium bromide (MTT) tests. Microglia were seeded at 100,000 cells per well in 96-well plates and treated with 0.5 µM, 1 µM, 5 µM or 50 µM empagliflozin in the presence or absence of 5 ng/mL LPS for 24 h. After washing, 100 µL of fresh medium supplemented with 25 µL MTT solution (1 mg/mL, Sigma-Aldrich) was added and cells were incubated for 2 h (37°C, 5% CO_2_). Subsequently, 100 µL solubilization solution (20% (*w/v*) SDS, 2.5% (*v/v*) 1 N HCL, 2.5% (*v/v*) acetic acid (80%) in 50% (*v/v*) DMF, pH 2) was added. Cells were incubated for 2 h (37 °C, 5% CO_2_) followed by the measurement of produced formazan at 550 nm using an automated plate reader (EAR340, ATTC, SLT).

The impact of microglia conditioned medium on neuronal cell viability was assessed as follows: microglia were cultivated under different conditions (control, 5 ng/mL LPS, 50 µM empagliflozin, 50 µM empagliflozin + 5 ng/mL LPS) as indicated above and supernatant was retrieved after 24 h (conditioned medium). SH-SY5Y cells (CRL-2266, ATCC) were maintained in DMEM/-ham’s F12 (1:1) medium (complemented with FCS, penicillin and streptomycin) and seeded at 20,000 cells per well in a 96-well plate. After 24 h the medium was replaced by microglia conditioned medium under different conditions (control, LPS, empagliflozin, empagliflozin + LPS) and cells were allowed to grow for a further 24 h. Cell viability was assessed using WST-1 assay (Roche, Mannheim, Germany) performed according to the manufacturer’s recommendations.

### 2.4. Measurement of Nitric Oxide Production

The level of nitric oxide production was determined indirectly by measuring nitrite accumulation in cell supernatants using Griess reagent (1% sulfanilamide and 0.1% N-(1-naphthyl)-ethylenediamine dihydrochloride in 5% H3PO4, Sigma-Aldrich). Primary microglial cells were seeded at 1,000,000 cells per well in 12-well plates. After stimulation for 24 h, 100 µL of each cell culture supernatant was mixed with 100 µL Griess reagent. After incubation for 15 min at room temperature, optical density (OD) was measured at 550 nm. Results were obtained from 12 independent experiments for each concentration, while measurements were performed in triplicate for each experiment.

### 2.5. Quantitative Real Time PCR

Microglia were seeded at 1,000,000 cells per well in 12-well plates and stimulated as described above for 24 h. Murine kidneys were retrieved from adult animals that were sacrificed for an unrelated project in agreement with the local Ethics Committee (242-70056/2015(91-7/15)) [19] and directly lysed in TRIZOL reagent (Invitrogen) using a homogenizator (Precellys Peqlab, Erlangen, Germany). Total RNA was isolated with the phenol chloroform extraction method using TRIZOL reagent according to manufacturer’s instruction. DNA was removed using RNAse-free DNAse I (Promega, Walldorf, Germany). After the addition of a random hexamer primer mix (Thermo Fisher Scientific, Karlsruhe, Germany), cDNA synthesis was performed using the RevertAid™ H Minus M-muLV reverse transcriptase (Thermo Fisher Scientific). RT-qPCR was performed using TaqMan Assay primers/probes (Thermo Fisher Scientific) or 5x HOT FIREPol^®^ EvaGreen^®^ qPCR Supermix (Solis BioDyne, Tartu, Estonia) (primers/probes references indicated in Table 1). *Gapdh*, *18s* and *Rps6* were used as housekeeping genes and internal controls. RT-qPCR was carried out on an ABI fast 7500 thermocycler (Applied Biosystems, Darmstadt, Germany). Relative changes in expression of the target gene in the treatment group related to the control were measured using the 2^−ΔΔCT^ method. All results are represented as a percent change of gene expression relative to LPS-stimulated microglial cells (for *Nos2*, *Il6*, *Il1b*, *Tnf*) or relative to unstimulated control (for *Nhe-1*), which were normalized to 100%. For the investigation of *Nhe-1*, *Sglt1* and *Sglt2* mRNA expression, the RT-qPCR products were run on a 2% agarose gel.

### 2.6. Enzyme-Linked Immunosorbent Assays

The cytokine concentration of IL6, IL1B, TNF and IL10 was measured in the cell culture supernatants using enzyme-linked immunosorbent assays (ELISAs). Microglia were seeded at 1,000,000 cells per well in 12-well plates and stimulated for 24 h. All ELISAs were carried out according to manufacturer’s instructions (BD OptEIA™ Rat IL-6 (IL6), BD Biosciences, Heidelberg, Germany; BD OptEIA™ Rat TNF (TNF), BD Biosciences; BD OptEIA™ Rat IL-10 (IL10), BD Biosciences; Rat IL-1 beta Tissue Culture Uncoated ELISA (IL1B), Invitrogen).

### 2.7. Western Blot

Western blot analysis was performed as previously described [20]. Microglia were seeded at 1,000,000 cells per well in 12-well plates. After stimulation for 30 min, microglia were lysed by adding lysis buffer (50 mM Tris (pH 7.5), 100 mM NaCl, 5 mM EDTA, 1% (*v*/*v*) Triton X-100, 2 mM sodium vanadate, 2.5 mM sodium pyrophosphate, 1 mM β-glycerol-phosphate and 1 mM phenylmethyl-sulfonylfluoride in acetonitrile). Protein quantification was carried out using a Pierce^TM^ BCA (bicinchoninic acid) protein assay kit (Thermo Fisher Scientific). Protein samples of 5 µg each were denaturized in SDS buffer for 5 min at 99 °C. Separation was performed on a 10% SDS gel and transferred onto a PVDF membrane (Roth, Karlsruhe, Germany). After blocking, immune detection was performed using an anti-phosphorylated ERK antibody (rabbit Phospho-p44/42 MAPK (ERK1/2), #4370, 1:1000, Cell Signaling Technology, Frankfurt am Main, Germany) or anti-ERK1/2 antibody (mouse Anti-MAP Kinase 2/Erk2, #05-157, 1:1000, Millipore, Darmstadt, Germany). Visualization was performed with HRP conjugated anti-rabbit/mouse secondary antibody (chicken anti-mouse IgG-HRP, sc-2962, 1:10,000, Santa Cruz Biotechnology, Dallas, USA, and goat anti-rabbit IgG-HRP, sc-2004, 1:10,000, Santa Cruz Biotechnology) and enhanced chemiluminescence (ECL WB detection reagent, Amersham Pharmacia Biotech, Amersham, UK). Beta-actin was used as the loading control (primary antibody: mouse β-actin (C4), sc-47778, 1:500, Santa Cruz; secondary antibody: Goat anti-Mouse IgG (H + L) Cross-Adsorbed, HRP, G-21040, 1:5000, Invitrogen). Fluorescence intensity was measured with a Fusion SL detection system (peqLab) running FusionCapt Advance solo 4 16.09b software (Vilber Lourmat, Eberhardtzell, Germany).

### 2.8. Immunofluorescence Staining of NFkB

Immunofluorescence staining was performed as previously described [21]. Microglia were seeded at 100,000 cells per glass cover slip. After stimulation for 30 min, cells were fixed and permeabilized with acetone. Free binding sites were blocked with bovine serum albumin (Serva, Heidelberg, Germany) for 30 min at room temperature. Microglial cells were incubated for 60 min with 50 µL of the primary anti-NFkB antibody (mouse NFkB p65 (F-6), sc-8008, 200 µg/0.5 mL, Santa Cruz), diluted at 1:50 in antibody diluent (Thermo Fisher Scientific). After washing, fluorescent conjugated secondary antibody (Alexa Fluor A 31,620, anti-mouse, Invitrogen) diluted at 1:700 was added, and cells were incubated for 45 to 60 min at room temperature. Nuclei were counterstained with Bisbenzimid H 33342 (Hoechst, Sigma-Aldrich) for 5 min. After mounting (ImmuMount, Thermo Fisher Scientific), analysis was performed using an inverted fluorescence microscope (Axiovert 200M, Carl Zeiss, Jena, Germany) and the software AxioVision (Carl Zeiss). Nuclear NFkB immunoreactive signal intensity was quantified using ImageJ 1.53q by creating a mask for the nuclear area (Hoeschst, blue) and measuring the mean grey values of the NFkB (green) channel corresponding to this nuclear area. Representative images of 3 independent experiments were used for quantification.

### 2.9. Statistical Analysis

Statistical analyses were performed using GraphPad Prism 9.31 software. The results of at least three independent experiments were analyzed using analysis of variance (ANOVA) followed by Dunnett’s post-hoc test (MTT test, WST-1 assay) or Tukey’s post-hoc test (NO release, qPCR, ELISA, Western blot, quantification of immunostaining). Results with *p* ≤ 0.05 were considered significant. All data are shown as mean values ± SEM (standard error of the mean).

## 3. Results

### 3.1. Empagliflozin Shows No Cytotoxic Effect on Primary Microglia up to a Concentration of 50 µM

Cytotoxic effects of empagliflozin on primary microglia were excluded via an MTT-assay. Microglia were stimulated with 0.5 µM, 1 µM, 5 µM and 50 µM empagliflozin in the presence or absence of LPS (5 ng/mL) or with LPS alone. The cell survival rate of an unstimulated control group was considered as 100%. Cell viability remained unchanged in the different groups after stimulation in comparison to control (Figure 1). In the subsequent investigations, 50 µM empagliflozin was used.

### 3.2. Empagliflozin Reduces Nos2 mRNA Expression in Activated Primary Microglia but Not NO Release

NO release is an important characteristic of activated microglial cells and a relevant pathological mechanism in neuroinflammation. The stimulation of primary microglia with LPS resulted in a significant increase in NO levels in the cell supernatant (Figure 2A). Incubation with 50 µM empagliflozin showed no influence on NO release after 24 h.

The inducible NO synthase (NOS2) is primarily responsible for NO synthesis in microglia. The influence of empagliflozin on *Nos2* mRNA expression was measured using qPCR. After 24 h stimulation with LPS, a significant increase in *Nos2* mRNA expression was observed (mean LPS Ct value: 24.71 ± 1.33; normalized to 100%). Co-treatment with 50 µM empagliflozin significantly reduced the LPS-stimulated *Nos2* mRNA expression by 53.93% (Figure 2B). Treatment with empagliflozin alone showed no significant effects on *Nos2* mRNA expression.

### 3.3. Empagliflozin Reduces mRNA Synthesis and Protein Production of Pro-Inflammatory Cytokines in Activated Primary Microglia

Neuroinflammation is associated with an increased expression of pro-inflammatory cytokines by microglia. The stimulation of primary microglia with LPS resulted in a significant increase in *Il6* (mean LPS Ct value: 26.38 ± 0.47), *Tnf* (mean LPS Ct value: 26.45 ± 0.59) and *Il1b* (mean LPS Ct value: 23.19 ± 0.44) mRNA expression (Figure 3A–C, normalized to 100%). Co-treatment with 50 µM empagliflozin significantly decreased the LPS-induced mRNA-expression of *Il6* by 65.52% (Figure 3A), *Il1b* by 45.48% (Figure 3B) and *Tnf* by 33.06% (Figure 3C) after 24 h. Treatment of primary microglia with empagliflozin alone did not affect the mRNA expression of *Il6*, *Il1b* and *Tnf* compared to the control group.

Further, the anti-inflammatory potential of empagliflozin was measured at the protein level by the quantification of IL6, IL1B and TNF concentrations in the cell culture medium via ELISA. The stimulation of primary microglia with LPS for 24 h significantly increased the protein expression of IL6 (2927 ± 542.5 pg/mL) and TNF (10,800 ± 4091 pg/mL), which was set at 100%. Co-treatment with 50 µM empagliflozin significantly decreased the LPS-induced protein release of IL6 (Figure 4A) and TNF (Figure 4B) by, respectively, 13.62% and 49.23%. The stimulation of primary microglia with empagliflozin alone had no influence on the release of IL6 and TNF proteins. The protein release of IL1B could not be detected in the supernatant using ELISA (data not shown).

### 3.4. Empagliflozin Reduces the IL10 Production in Activated Primary Microglia

The anti-inflammatory cytokine IL10 plays a major role in balancing inflammatory response. To investigate the effect of empagliflozin on IL10 protein release by primary microglia, the cytokine concentration in the cell culture medium was measured by ELISA. Stimulation with LPS resulted in a significant increase in IL10 release (1391 ± 158 pg/mL). Co-treatment with LPS and 50 µM empagliflozin for 24 h significantly decreased the IL10 protein release compared to the LPS-stimulated primary microglia by 22.97% (Figure 5). Incubation with 50 μM empagliflozin alone did not affect IL10 protein release in the cell culture supernatant of primary microglia in comparison to the control group.

The impact of microglia conditioned medium on neuronal cell viability was further evaluated. Although SH-SY5Y viability was slightly reduced after being cultured for 24 h with a conditioned medium of LPS-treated microglia, we failed to detect any significant impact of empagliflozin on these parameters (data not shown).

### 3.5. Empagliflozin Might Mediate Anti-Inflammatory Effects by Inhibiting the ERK1/2 and NFkB Signaling Pathway

The MAP kinases ERK1/2 are involved in the complex regulation of microglial inflammation. After the activation of this signaling pathway, ERKs MAP kinases are phosphorylated, which then migrate to the cell nucleus and influence the gene expression of inflammatory cytokines. To investigate the degree of activation of this signaling pathway, phosphorylated ERK1/2 (pERK1/2) and total ERK1/2 were measured by Western blot after stimulation for 30 min. Co-treatment with LPS and 50 µM empagliflozin significantly decreased the pERK/ERK ratio by 25.76% in comparison to LPS-treated cells (Figure 6). No significant alterations of ERK phosphorylation were observed after stimulation with empagliflozin alone, as compared to the control group.

In addition to the MAP kinases, the transcription factor NFkB is an important mediator of inflammatory processes in microglia. The translocation of NFkB to the nucleus was assessed using immunocytochemistry. Whereas NFkB was predominantly observed in the cytoplasm of untreated cells, NFkB was mainly found in the nucleus of cells treated with LPS (Figure 7). The occurrence of NFkB within the cell nucleus was significantly reduced after co-treatment with LPS and 50 µM empagliflozin in comparison to the LPS-stimulated primary microglia. The distribution pattern of NFkB in the microglia incubated with empagliflozin alone was similar to the distribution pattern observed in the control group.

### 3.6. The Anti-Inflammatory Properties of Empagliflozin Might Be Mediated via the Sodium/Hydrogen Exchanger NHE-1 Rather Than SGLT2

Empagliflozin is primarily an inhibitor of SGLT2. SGLT1 (solute carrier family 5 member 1, SLC5A1) and the sodium/hydrogen exchanger NHE-1 (solute carrier family 9 member A1, SLC9A1) are also described as potential off-targets of empagliflozin.

The mRNA expression of *Sglt1*, *Sglt2* and *Nhe-1* in primary microglia was assessed using RT-qPCR after 24 h stimulation with empagliflozin (50 µM) and LPS (5 ng/mL) alone or in combination. The mRNA expression of these target genes in murine kidney was used as the positive control. Whereas *Nhe-1* was expressed at high levels in microglia (mean LPS Ct value: 27.05 ± 0.90; Figure 8A,B), an expression of *Sglt1* (Figure 8A) and *Sglt2* (Figure 8A) was barely detectable. The treatment of primary microglia with empagliflozin or LPS, alone or in combination, did not affect the *Nhe-1* mRNA expression significantly.

## 4. Discussion

Neurodegenerative disorders and T2DM have developed into a burden for our society. It is noteworthy that patients presenting with T2DM have an increased risk of developing neurodegenerative disorders, including PD or AD [22,23,24]. Although some patients presenting with both neurodegenerative disorders and T2DM may be on SGLT2-inhibitor medication, little is known regarding the impact of such drugs on neurobiological outcome. Recent studies indicate that the antidiabetic drug empagliflozin exerts not only a hypoglycemic effect, but also an inflammation-modulating influence on various cell types, including macrophages [25,26]. Based on these findings, our study aimed to investigate the effect of empagliflozin on LPS-mediated reactive primary microglia.

The pathomechanisms underlying neurodegenerative disorders are thought to rely in a large part on the dysbalanced regulation of pro- and anti-inflammatory cytokine release by microglia [2,27,28,29].

Pro-inflammatory cytokines, including IL6, TNF and IL1B, are not only involved in the direct mediation of neurotoxic effects, but also lead to an increased permeability of the blood–brain barrier and further recruitment of immune cells [30,31]. There is striking evidence that levels of pro-inflammatory cytokines are elevated in the blood and brains of patients with neurodegenerative disorders, which appear to correlate with the severity of symptoms [32,33,34]. Although a chronic low-level expression of TNF has been shown to exert neuroprotective effects in mice exposed to intrastriatal 6-hydroxydopamine, a chronic exposure of the CNS to high concentrations of IL1B or TNF leads to microglial activation and irreversible damage to dopaminergic neurons [35,36]. While previous work on macrophages has shown empagliflozin’s potential to reduce the expression of IL6, IL1B and TNF, the effect on cytokine release by primary microglia has not yet been investigated [25,37]. Our study demonstrated that empagliflozin is able to ameliorate the LPS-induced mRNA expression of *Il6*, *Tnf* and *Il1b* in primary microglia. Further, we demonstrated a reduced protein release of IL6 and TNF after empagliflozin stimulation. These results are consistent with the reduced microglia burden in AD-T2D mice after empagliflozin treatment, and suggest that microglia may represent a direct target for empagliflozin in this context [17]. However, we failed to detect any IL1B release in our cellular model, indicating that the effects of empagliflozin in this context may be limited to the regulation of pro-*Il1b* [38].

Similar conflicting results were obtained regarding NO production. Indeed, in our study, we observed that empagliflozin inhibits the LPS-induced *Nos2* mRNA expression, but had no effect on NO release. Analogous results have been observed in rat myocardium, where empagliflozin has been shown to reduce *Nos2* expression, while the activity of *Nos1* and *Nos3* was increased, which led to higher NO levels [39]. However, LPS-induced NO production in glial cells has been shown to depend almost exclusively on NOS2 activity, suggesting that the observed effects of empagliflozin may be insufficient to restrain NOS2-mediated NO production in our model [40]. Nonetheless, the inhibition of NOS2 may also exert beneficial neuroprotective effects, independently of NO-production, through the inhibition of oxidative species formation [39,41].

The anti-inflammatory cytokine IL10 is characterized as a regulating cytokine, which initiates the termination of an inflammatory response to prevent excessive inflammation [27,42]. In vivo models of PD showed a reduction in microglial activation and reduced dopaminergic neuronal cell death after the administration of IL10 [43,44]. Therefore, the induction of IL10 release might be beneficial in several neurodegenerative diseases, where inflammation is harmful [45]. Our study demonstrates that the LPS-induced IL10 release was significantly reduced by co-treatment with empagliflozin. Thus, the inhibitory impact of empagliflozin on the release of pro-inflammatory mediators observed in this work appears to affect the release of the anti-inflammatory cytokine IL10 in a similar way. These results are in line with in vivo experiments in mice, which showed a significant reduction in serum IL10 levels after six weeks of empagliflozin intake [46]. In contrast, patients with T2DM, who received empagliflozin for 24 weeks, presented with significantly higher serum levels of IL10 in comparison to patients, who did not receive the SGLT2 inhibitor [47]. Therefore, the regulation of IL10 under a pathological state certainly involves more complex regulatory mechanisms, which remain to be identified.

The production of pro-inflammatory cytokines is regulated in part by the ERK1/2-MAP-kinases pathway, which itself regulates the activation of the transcription factor NFkB [48,49]. Contradictory results have been described regarding the impact of empagliflozin on the ERK1/2-MAP-kinases pathway in different tissues [50,51]. Nonetheless, empagliflozin has been shown to inhibit NFkB activation in the kidney and in the liver, as well as in endothelial cells and macrophages [37,52,53,54,55,56]. Although further studies are required to fully decipher the impact of empagliflozin on these pathways in the context of neuroinflammation, our study provides the first evidence that empagliflozin may similarly inhibit the ERK1/2-MAP-kinases pathway in microglia.

Considering that SGLT1 and SGLT2 are expressed at very low levels, if at all, in primary microglia, it is likely that the observed anti-inflammatory effects of empagliflozin may be mediated through unspecific, or off-target, properties, which would need to be deeply addressed in future investigations. One such off-target may be NHE-I, which appears to be expressed at significant levels in microglia, as shown in this study and previous work [57]. Indeed, NHE-I has been proposed to mediate empagliflozin’s effects on cardiac cells, although such direct effects are still at the center of debates in the literature [58,59,60,61]. It is noteworthy that NHE-1 has been shown to be involved in LPS-induced inflammatory effects in microglia, suggesting that its potential inhibition through empagliflozin may indeed play a role in the anti-inflammatory effects observed in our study [57].

SGLT2 has been shown to be expressed in different parts of the brain, including the amygdala and the hypothalamus, and seems to be predominant in blood vessels of the blood–brain barrier, although little is known about its expression at the cellular level [16,62]. It appears evident that the impact of empagliflozin on neuroinflammation in vivo may involve complex cellular processes, which certainly may be mediated by additional cell types, including endothelial cells, astrocytes, as well as neurons themselves, and which should be investigated in future studies [62].

Other SGLT2 inhibitors, including canagliflozin and dapagliflozin, have found their way to the clinic, and several additional compounds are currently in preclinical test phases. Whether these inhibitors similarly target microglia cells is still unknown, but anti-inflammatory properties have been proposed both for canagliflozin and dapagliflozin [63,64], and dapagliflozin may similarly have off-target effects on NHE-1 [65].

Recent studies have demonstrated potential cerebral anti-oxidative [18,66], vascular [17] and blood–brain barrier protective [67], as well as neuroprotective [18,66], properties of empagliflozin [16,68]. Empagliflozin has been proposed to cross the blood–brain barrier in mice (brain/serum ratio of 0.5); however, data for humans do not yet exist [69]. In line with these studies, our results indicate that empagliflozin may exert slight direct anti-inflammatory properties in microglia, as characterized by the reduced expression of the pro-inflammatory mediators *Nos2*, IL6, TNF and *Il1b*. Taken together, our results suggest that empagliflozin may not exacerbate the inflammatory reaction, but rather exert beneficial effects in patients presenting with neurodegenerative disorders and concomitant T2DM. Although a clinical randomized trial of the SGLT2 inhibitor dapagliflozin in patients with AD is currently being conducted (NCT03801642), to our knowledge, similar studies have not yet been performed for empagliflozin [16]. Our results underline the need for further studies addressing potential beneficial effects of empagliflozin in the context of neuroinflammation, in particular in patients with neurodegenerative disorders.

## Figures and Tables

**Figure 1 cells-11-03107-f001:**
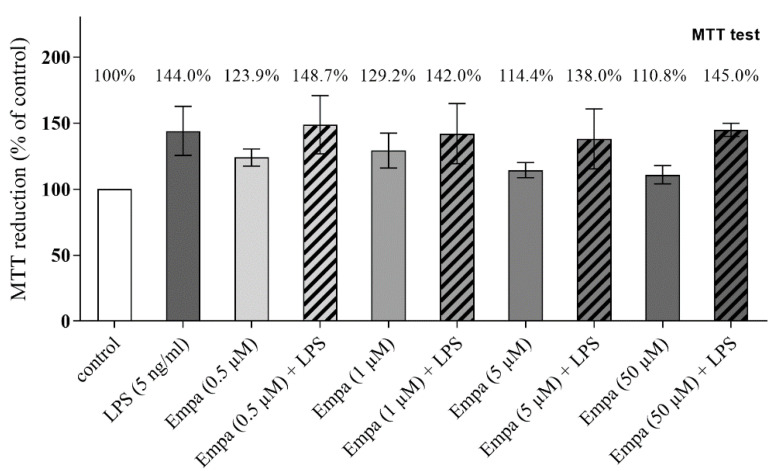
The MTT test was performed to determine the cell viability of primary microglia after treatment with empagliflozin (Empa) at increasing concentration (0.5, 1, 5, 50 µM) alone or in combination with LPS (5 ng/mL). Cell viability remained unchanged in the different groups after stimulation in comparison to control (one-way ANOVA, followed by Dunnett’s test; F-value = 1.286; *n* = 4).

**Figure 2 cells-11-03107-f002:**
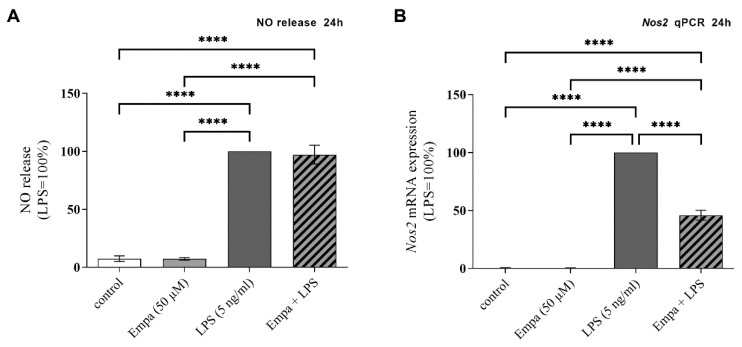
LPS significantly increased NO release (**A**) and *Nos2* mRNA expression (**B**) in primary microglia. Treatment with 50 µM empagliflozin showed no influence on NO release (**A**), while LPS-induced *Nos2* mRNA expression was significantly decreased after 24 h (**B**) (asterisks indicate all significant differences; **** *p* ≤ 0.0001; one-way ANOVA, followed by Tukey test; F-value = 1797 (**A**), 1438 (**B**); *n* ≥ 3).

**Figure 3 cells-11-03107-f003:**
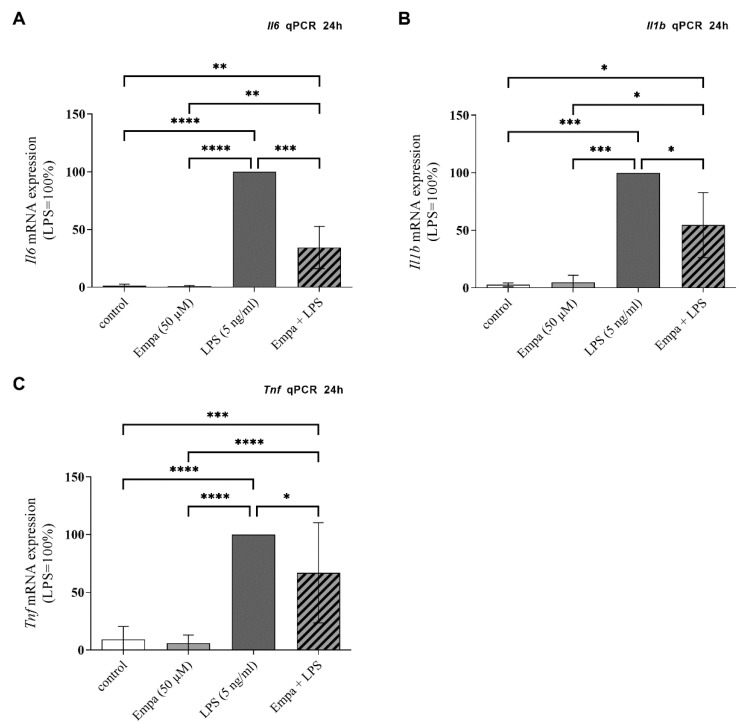
The mRNA expression of pro-inflammatory cytokines by primary microglia was measured using qPCR after stimulation for 24 h. Empagliflozin significantly reduced the LPS-induced mRNA expression of *Il6* (**A**), *Il1b* (**B**) and *Tnf* (**C**). Stimulation with empagliflozin alone did not affect the mRNA expression in comparison to the control (asterisks indicate all significant differences; * *p* ≤ 0.05, ** *p* ≤ 0.01, *** *p* ≤ 0.001, **** *p* ≤ 0.0001; one-way ANOVA, followed by Tukey test; F-value = 77.41 (**A**), 30.69 (**B**), 32.72 (**C**); *n* ≥ 3).

**Figure 4 cells-11-03107-f004:**
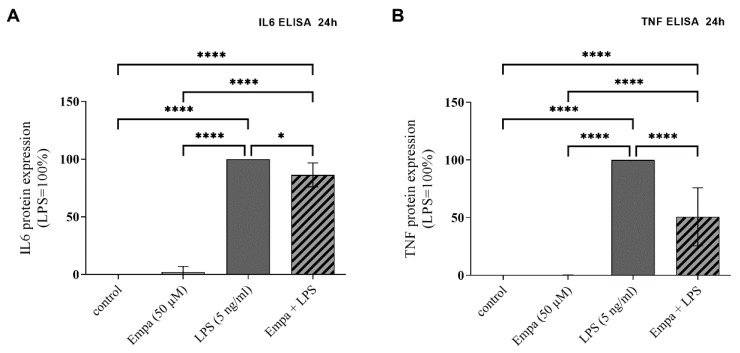
Cytokine release of IL6 and TNF by primary microglia was measured using ELISA after stimulation for 24 h. Empagliflozin significantly reduced the LPS-induced protein expression of IL6 (**A**) and TNF (**B**). Stimulation with empagliflozin alone did not affect the protein expression in comparison to control (asterisks indicate all significant differences; * *p* ≤ 0.05, **** *p* ≤ 0.0001; one-way ANOVA, followed by Tukey test; F-value = 351.2 (**A**), 87.39 (**B**); *n* ≥ 4).

**Figure 5 cells-11-03107-f005:**
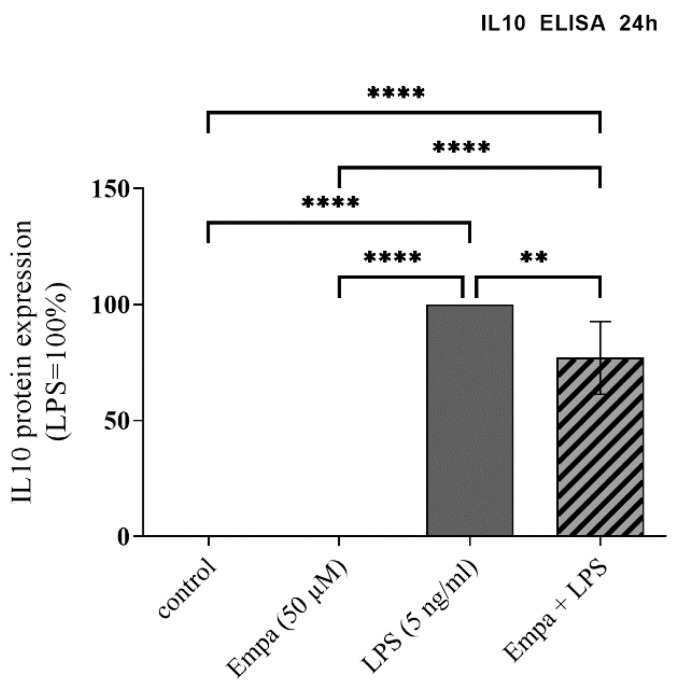
Protein expression of the anti-inflammatory cytokine IL10 was measured using ELISA after stimulation for 24 h. Empagliflozin significantly reduced the LPS-induced protein expression of IL10, while stimulation with empagliflozin alone showed no influence in comparison to the control (asterisks indicate all significant differences; ** *p* ≤ 0.01, **** *p* ≤ 0.0001; one-way ANOVA, followed by Tukey test; F-value = 218.5; *n* = 5).

**Figure 6 cells-11-03107-f006:**
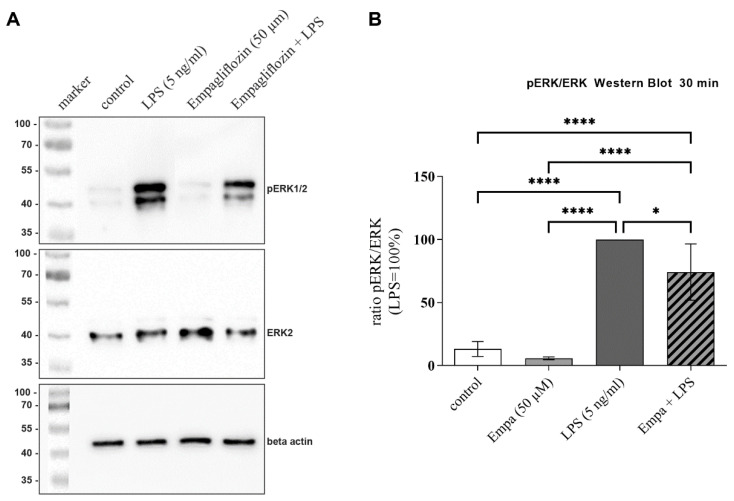
Phosphorylation of ERK1/2 was measured using Western blot after stimulation for 30 min (**A**). Beta-actin was used as the loading control. Bands are identified by their molecular mass based on markers. Quantification of band intensities of ERK2 and pERK1/2 and calculation of pERK/ERK ratio (**B**). LPS activation of microglia caused a strong increase in pERK1/2 levels. Through simultaneous incubation with empagliflozin, LPS-induced phosphorylation of ERK1/2 was significantly decreased by 25.76% (asterisks indicate all significant differences; * *p* ≤ 0.05, **** *p* ≤ 0.0001; one-way ANOVA, followed by Tukey test; F-value = 63.75; *n* = 4).

**Figure 7 cells-11-03107-f007:**
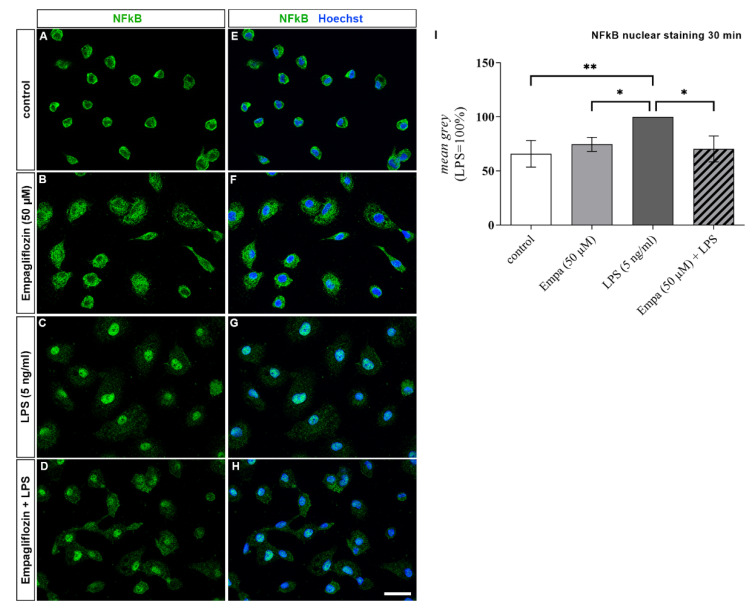
Representative images of immunofluorescence staining showing the distribution pattern of NFkB (green) in rat primary microglia after 30 min of stimulation. Nuclei were counterstained with Hoechst (blue). Nuclear NFkB signal intensity was quantified using ImageJ (**I**). Unstimulated control cells showed predominately cytoplasmatic NFkB staining (**A**,**E**), whereas LPS (5 ng/mL) stimulation resulted in a significant NFkB translocation into the nuclei (**C**,**G**,**I**). Co-treatment with LPS and empagliflozin (50 µM) significantly reduced NFkB translocation (**D**,**H**,**I**). Distribution pattern of microglia stimulated with empagliflozin alone (**B**,**F**,**I**) was similar to control cells (scale bar = 20 µM; asterisks indicate all significant differences; * *p* ≤ 0.05, ** *p* ≤ 0.01; one-way ANOVA, followed by Tukey test; F-value = 8.266; *n* = 3).

**Figure 8 cells-11-03107-f008:**
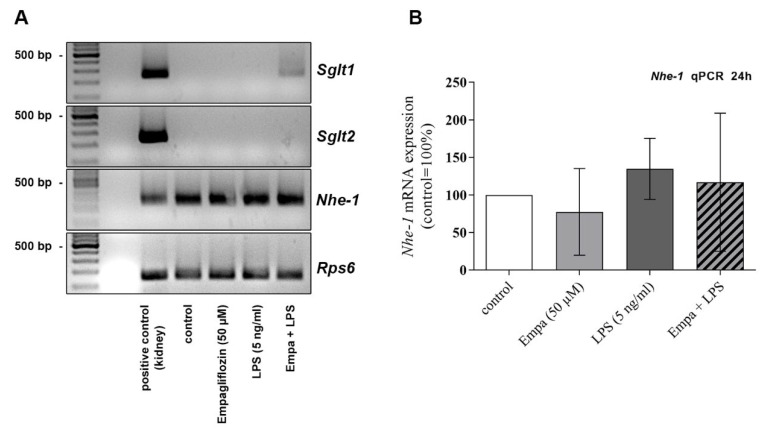
*Sglt1*, *Sglt2* and *Nhe-1* mRNA expression in primary microglia was measured using RT-qPCR. Kidney was used as positive control. Migration profiles of RT-qPCR products on agarose gel (**A**). *Nhe-1* was expressed at high levels in primary microglia (**A**,**B**) whereas *Sglt1* and *Sglt2* (**A**) expression was barely detectable. Stimulation for 24 h with empagliflozin and LPS, alone or in combination, did not alter the *Nhe-1* mRNA expression (**B**, one-way ANOVA, followed by Tukey test; F-value = 1.242; *n* = 6).

**Table 1 cells-11-03107-t001:** Primers/probes references.

Gene		Sequence (5′-3′) or Reference
*Nos2*	Thermo Fisher Scientific	Rn00561646_m1
*Il6*	Thermo Fisher Scientific	Rn00561420_m1
*Il1b*	Thermo Fisher Scientific	Rn00580432_m1
*Tnf*	Thermo Fisher Scientific	Rn99999017_m1
*18s*	Thermo Fisher Scientific	Hs99999901_s1
*Gapdh sense*	Eurogentec, Köln, Germany	CAGCAAGGATACTGAGAGCAAGAGA
*Gapdh antisense*	Eurogentec	CGATGGAATTGTGAGGGAGATG
*Gapdh probe*	Eurogentec	AGGAGTCCCCATCCCAACTCAGCCC
*Nhe-1 sense*	Sigma-Aldrich	GTACGCACACCCTTCGAGAT
*Nhe-1 antisense*	Sigma-Aldrich	CAGAGGCAGGAAGTAGCCTG
*Sglt1 sense*	Sigma-Aldrich	CCAGTGGGCTGTACCAACAT
*Sglt1 antisense*	Sigma-Aldrich	ATGCCAATCAGCACGAGGAT
*Sglt 2 sense*	Sigma-Aldrich	GCGTATTTCCTGCTGGTCATT
*Sglt 2 antisense*	Sigma-Aldrich	GAGGAGCAACACCACAAAGAG
*Rps6 sense*	Sigma-Aldrich	CCAAGCTTATTCAGCGTCTTGTTACTCC
*Rps6 antisense*	Sigma-Aldrich	CCCTCGAGTCCTTCATTCTCTTGGC

## Data Availability

Not applicable.

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
