# Peer review of "Anti-Inflammatory Properties of the SGLT2 Inhibitor Empagliflozin in Activated Primary Microglia"

_cells, 2022, doi:10.3390/cells11193107_

Round 1
Reviewer 1 Report
In this context, the authors investigated the effect of empagliflozin on activated primary microglia. There are some problems with this article.
1. Since the empagliflozin reduced both the expression of pro-inflammatory mediators and anti-inflammatory mediators in LPS-activated primary microglia,I think its final effect requires further experiments such as culturing neuron with conditioned medium.
2. 50 μM empagliflozin significantly reduced the LPS-stimulated iNOS mRNA expression. However, treatment with 50 μM empagliflozin showed no influence on NO release (Fig.2). Although the author gives some explanations, the explanations are not very reasonable.
3. Figure 6A needs to provide a clearer picture.
4. Figure 7 requires a statistical result.
Reviewer 2 Report
The study from Heimke, Lenz et al., intends to investigate the role of glucose transporter inhibitor Empaglifozin in the activated primary microglial model, as a possible inhibitor of neuroinflammation. The study is well-conducted and brings relevant information about how drugs used in diabetes mellitus type 2 and glucose control can have meaningful beneficial effects on neuropathological conditions where inflammation is prominent. Some small points need to be addressed before publication.
Material and Methods:
Isn’t 105 cells too much for proper confluence in 96-well plates? Same for 106 in 12-well plates.
Two-way ANOVA should be more appropriate in your statistical analysis to check for drug interaction (which seems to be the case in your results)
Why wasn’t a loading control used together with p-ERK and total ERK ½?
Results:
The title “Empagliflozin reduces NO release by activated primary microglia” does not reflect your data. Despite having a reduction in mRNA expression of iNOS, you do not see an altered LPS-induced NO release after empaglifozin (figure 2A).
Does SGLT2 have a significant expression in microglia? The subject is not presented in the introduction section. It would be interesting to see, perhaps through imaging techniques, the expression of these transporters in microglia.
Because IL-1beta has a very different synthesis pathway of IL-6 nd TNF-α (through inflammasomes) in microglia, it would be interesting to see its protein levels by ELISA as well.
I believe it would be important to quantify colocalization of NFkB and nuclei. Specially because your photomicrographs give me the impression that you have more NFkB staining in empaglifozin alone-treated groups (or at least that they are more diffuse in the cytoplasm). A quantification would easily solve that impression and avoid misinterpretations.
Discussion:
Because the authors bring the constant discussion about neurodegenerative disorders, it would be interesting to see if the conditioned medium of empaglifozin-treated microglia would have a lessened impact on neuronal viability.
I understand empaglifozin is effective in your model and the results are evident. But I wonder if there are no other SGLT transporters in microglia that could have even more significant effects. This information should be added in your discussion section.
Reviewer 3 Report
Microglial cells, the resident immune cells of the brain, play important roles and react to any insults in the brain (cell death, infection, injury …). Heimke M. and collaborators describe anti-inflammatory effects of empagliflozin on rat primary microglia.
The authors treated their primary cultures with LPS at 5 ng/ml to induce microglial reactivity and to mimic a neuroinflammatory state. In this condition, a pre-treatment with Empagliflozin decreases the LPS-induced pro/anti-inflammatory gene overexpression (Il1b, Il6, Il10, Nos2, Tnf) after 24h. The authors also described a decrease of the release of pro-inflammatory mediators (IL-6 and TNFa). These effects might be mediated by inhibition of the ERK1/2 and NFkB pathways.
The data have been generated following real-time PCR, western-blots and ELISAs experiments coupled to nitrite accumulation quantification and ICC analyses.
As a summary, the authors provide new data to reduce microgliosis, a mechanism that is observed during neuroinflammation and that might be of importance in many neurodegenerative diseases. I would like to point out that very few papers described anti-inflammatory effects of Empagliflozin on the CNS and mainly on glial cells. Even if the authors used an over-simplified model of neuroinflammation, this manuscript brings new data in the field. Furthermore, the manuscript is concise and well written.
Specific comments for the authors are listed below.
Major comments
1) NO release and ELISAs
Regarding the LPS treatment, what are the concentrations (in mM or pg/ml) corresponding to the 100% values in term of nitrite production (Fig. 2A), and IL-6 / TNFa / IL-10 releases (Figs. 4 and 5)? Please indicate the real values in the text.
2) Western-blotting results
- I would really appreciate to have the migration profiles of the protein of reference that authors used in their experiments in ctrl, LPS, Emp. and LPS + Emp. conditions. This reference protein must also be mentioned in the text.
- Empagliflozin has been described in the literature to modulate ERK1/2, but also p38 MAPK. This characteristic is important since p38 MAPK activity is also linked to neuroinflammation and glial cells (mainly astrocytes and microglia). Why do the authors only focus on ERK1/2 and not on both? Is p38 modulated in the same way as ERK1/2 under these conditions?
3) qPCR data
- In order to properly estimate the microgliosis intensity level, it would be valuable to mention in the text the corresponding Ct values for Il6, Il1b, Tnf and Nos2 expression at least for the LPS condition.
- How can the authors be sure that the Il1b mRNA expression reflects the mature IL1b and not the pro-IL1b? Do they have performed an IL1b ELISA to confirm this point?
4) Microglial morphology
In non-treated conditions, microglial cells have a ramified morphology contrary to what is shown in Figure 7. Do the authors have an explanation?
Minor comments
1) Treatments of microglia
- Please mention in the text when microglial cultures wee treated after plating. After 24h or more?
- Regarding the cytotoxicity tests (Section 2.3), is it correct that cells are incubated at 37°C and 8.3% CO2? If so, authors should explain why such CO2 value?
2) Gene names
Based on the current nomenclature, please correct gene names (Il1b, Il6, Nos2, Tnf, Gapdh) in all the document
3) Western blots
What are the protein quantities (in mg) that have been used for WB experiments. This must be mentioned in the Mat/Methods Section.
4) Secondary antibody for ICC
Please verify the anti-mouse antibody used for NFkB staining (line 171)
5) Empagliflozin concentrations
Please indicate Empagliflozin concentrations (line 93) as mentioned in the different figures in mM
6) Typos:
Please check typos in all the document (lines 40, 90, 150, 353)
Reviewer 4 Report
Dear Authors,
The manuscript titled "Anti-inflammatory properties of the SGLT2 inhibitor Empagliflozin in activated primary microglia" is very interesting and well written.
However, I have some concerns and suggestions in order to improve the quality of your paper.
MAJOR concerns:
1- Page 2, lines 94-97: The Authors treated the cells in co-stimulation. However, they stated that the Empaglifozin was added 30 min. before adding LPS. The question is: Did The Authors remove the empaglifozin during the LPS treatment or leave it in the medium during the LPS treatment? Furthermore, adding a stimulus 30 min before is not considered a co-stimulation but a pre-treatment. Indeed, in this way, the cells are able to react with the first stimulus and then with LPS.
2-Page 2, line 96-97: Why different time of stimulation (30 min., 60 min., 24 hours) for different experiments (ELISA, immunofluorescence, western blotting)? Please explain.
3- Page 4, line 166: immunofluorescence staining for NFKB: The Authors quoted that the stimulation was 30 min. At page 2, lines 96, they quoted that the stimulation was 60 min. Please explain. Moreover, please add the pre-stimulation time for empaglifozin.
4-Page 4, line 182: The data are shown considering the control as 100% (for MTT assay), and LPS treatment as 100% (for the other experiments). Please, use the same experimental point as 100% (control or LPS).
5- Page 9, Figure 7. I suggest the Authors to split the blue (DAPI) and green (NFkB) channels in order to better visualise the nuclear translocation.
6- At least one error bar is missing in all the graphs. Figure 1: MTT test (control has no error bar); All the other Figures are missing the error bar in the LPS experimental point. Please explain.
7 - Figure 1: please add the p value if a significant difference is present (for example control vs. Empa 50µM + LPS).
8- Figure 2, panel B: Is significantly different the Empa + LPS experimental point vs. LPS alone? The same question is for Figure 3, panels B and C; Figure 4, panel A; Figure 5; Figure 6, panel B Please, add the error bar to LPS as required at point 6.
9- Did the Authors perform experiments concerning also other oxidative stress markers (such as ROS, cytochrome C, Nrf2 translocation)? Indeed, since it has been reported that LPS induces microglia oxidative stress (10.3389/fncel.2020.00142) and different natural compounds are able to ameliorate such a situation (PMID: 32655777), it should be fine to highlight also the role of empagliflozin on oxidative stress reduction, as reported in leukocytes (10.3390/jcm8111814).
9- Did the Authors perform experiments with other SGLUT2 inhibitors such as canagliflozin, pagliflozin or ertugliflozin?
10- Did the Authors perform experiments on other cell types such as astrocytes?
MINOR concerns:
1- Page 3, line 101: MTT assay, please add the company name and place.
2-Page 3, line 103: LPS was abbreviated before. Please, remove the full-length word.
3- Page 2 (line 90), Page 3 (line 105), Page 3 (line 115): Please do not begin a sentence with a number.
4- Table 1: TNF-a should be replaced with the greek alphabet.
5- Figure 6, panel A: Please, add the molecular weight for western blotting bands.
Round 2
Reviewer 1 Report
Some suggestions are as follows:
1. For all data statistics, a point graph is more intuitive than a histogram and should be recommended. Setting the LPS group to 100% is incomprehensible, as are the difference labels a, b, and c.
2. In Figure 1, the statistical difference between some groups such as the LPS group (144.0%) and the control group needs to be re-determined and the solvent control group needs to be done.
3. Besides IL0, more anti-inflammatory factors should be detected.
4. In Figure 6A, the bands of ERK2 are poor quality.
5. Experiments with IbaI staining to confirm the purity of microglia should be required. The microglia did not appear to be quiescent in Figure 7A.
6. It is unreasonable that IL1B in the supernatant of microglia after LPS stimulation could not be detected by ELISA.
7. The anti-inflammatory properties of empagliflozin may not be mediated through SGLT2, but more evidence is needed to confirm whether it is mediated through NHE-1.
Reviewer 3 Report
The document has been modified in accordance with reviewers' comments. As an additional minor comment, check NFkB's writing again.
Reviewer 4 Report
Dear Authors,
The revised version of your paper has been largely improved and the paper should be considered for publication.
